

# Decomposition and scenario analysis of agricultural carbon emissions in Heilongjiang, China

Luyang Zhang[1,2], Junyan Chu[1], Haiyang You[1] and Zhihua Liu[1]

[1] School of Resources and Environment, Northeast Agricultural University, Harbin, Heilongjiang, China
[2] Joint Laboratory of Agriculture coping with Climate Change of China Meteorological Administration (CMA) and China Agricultural University (CAU), Beijing, China

## ABSTRACT

**Background**. As a key agricultural region in China, Heilongjiang Province has experienced significant carbon emissions over the past few decades. To understand the underlying factors and future trends in these emissions, a comprehensive analysis was conducted from 1993 to 2030.

**Methods**. The agricultural carbon emissions from 1993 to 2020 were estimated using the emission factor method. To analyze the influencing factors and future trends of these emissions, the study employed the Logarithmic Mean Divisia Index (LMDI) and integrated it with the Stochastic Impacts by Regression on Population, Affluence, and Technology (STIRPAT) model.

**Results**. Results showed that (1) the agricultural carbon emissions in Heilongjiang were primarily driven by rice cultivation, followed by fertilizer production and irrigation electricity. (2) The economic and labor structure effects were the main driving factors of agricultural carbon emissions, while the population, demographic, and intensity effects were the main inhibitors. (3) Agricultural carbon emissions in Heilongjiang Province peaked in 2016 with 69.6 Mt $CO_2$-eq and could subsequently decline by −3.92% to −4.52% between 2020 and 2030 in different scenario simulations. In the future, Heilongjiang Province should prioritize the reduction of agricultural carbon emissions from rice production. Adjusting the planting structure, managing the layout of rice paddies, and promoting the cultivation of dry rice varieties would significantly contribute to mitigating agricultural carbon emissions.

# INTRODUCTION

Human activities are the primary contributor to climate change (*IPCC, 2021*). Agricultural carbon emissions are intricately linked to climate change because agricultural practices generate carbon dioxide and constitute a significant source of non-carbon dioxide greenhouse gas (GHG) emissions (*Melillo et al., 2002*). The Sixth Climate Assessment Report by the United Nations Intergovernmental Panel on Climate Change (IPCC) revealed that anthropogenic GHG emissions from the agriculture and land use sector (AFOLU) contributed 13–21% of the total GHG emissions during 2010–2019 (*IPCC,*

Corresponding authors
Luyang Zhang, zly@neau.edu.cn
Zhihua Liu, zhihua-liu@neau.edu.cn

*2022*). Furthermore, agricultural production is responsible for average annual emissions of $CH_4$ and $N_2O$ amounting to $157 \pm 47.1$ Mt and $6.6 \pm 4.0$ Mt, respectively (*IPCC, 2022*). As a major agricultural nation, China has seen a 1.6-fold increase in grain production over the past two decades (*NBS, 2022*). However, this surge in agricultural output has been accompanied by increased carbon emissions, primarily attributed to the mechanization of farming practices and the application of fertilizers. Agriculture accounts for 24% of total GHG emissions in China (*Shi et al., 2023*), the dual challenges of emissions reduction and food security are paramount. In response, China escalated its nationally determined contributions, pledging to reach peak carbon emissions before 2030 and achieve carbon neutrality by 2060 (*UNFCCC, 2021*). Each province planned its own emission reduction strategy. As the top grain producer in China, Heilongjiang Province faces the dual challenges of neutralizing agricultural carbon and safeguarding food production. Therefore, to provide scientific guidance for the future, it is important to identify the changing characteristics of agricultural production in Heilongjiang over time and analyze the elements that influence it.

Currently, the estimation of carbon emissions from the agricultural sector is well-established. The IPCC carbon emissions coefficient method is the most popular for estimating national and regional agricultural carbon emissions. This method is convenient for calculating agricultural activities in terms of carbon emissions based on relevant coefficients. *Wu et al. (2023a)* have estimated the total carbon emissions from crop production in China by IPCC carbon emissions coefficient method and observed it peaked at 262.65 Mt in 2015. However, the selection of coefficients in a few studies may not have been appropriate, leading to errors in assessing carbon emissions. *Du et al. (2019)* and *Tian & Zhang (2013)* chose the carbon emission coefficient for fertilizer production based on a study from the United States, which may have resulted in a low assessment (*Shan et al., 2023*). Therefore, the coefficients based on the local situation in China were selected for a more accurate assessment. Additionally, studies have assessed carbon emissions from plowing and agricultural diesel use (*Ding et al., 2019*; *Huang & Zhang, 2022*). The carbon emissions from plowing are from the use of diesel in machinery, which is double counting, and only carbon emissions from diesel must be assessed.

Further studies to clarify the influencing factors and project changing trends in agricultural carbon emissions could help develop targeted carbon reduction strategies. The Logarithmic Mean Divisisa Index (LMDI) method has been widely used to identify the factors influencing carbon emissions owing to its convenient analysis process and data availability (*Ang, 2015*). *Li, Bai & Xiao (2017)* decomposed agricultural carbon emissions in Heilongjiang, China, and observed that, except for the carbon emission intensity effect, the effects of agricultural income, employment structure, and population acted as positive driving factors of agricultural carbon emissions during 1996–2013. However, agricultural carbon emissions in Heilongjiang have declined in recent years (*Wu et al., 2023a*). In addition, the population of Heilongjiang has decreased by 18.5% over the last decade (*NBS, 2022*). The relationship between the changes in these relevant factors and agricultural carbon emissions in Heilongjiang deserves further study.
Similar to the LMDI method, the Stochastic Impacts by Regression on Population, Affluence, and Technology (STIRPAT) model is another useful tool for estimating the driving forces of carbon emissions (*Rosa & Dietz, 1998*; *York, Rosa & Dietz, 2003*). The LMDI is mostly used as a retrospective tool, such as for the impact of structural changes, whereas the STIRPAT model can predict environmental impacts based on key driving forces. These key forces reflect the policy implementation well. Thus, the advantage of the STIRPAT model is its ability to provide recommendations for policy improvements, together with effective predictions. The STIRPAT model has been used in many studies to examine the impact factors of carbon emissions at the national or regional level (*Wang et al., 2013*; *Wu et al., 2021*) and to predict carbon emission trends in particular industry sectors (*Vélez-Henao, Vivanco & Hernández-Riveros, 2019*). Chinese research has estimated the mining industry (*Wei et al., 2023a*), building sector (*Zhu et al., 2022*), and household carbon emission factors (*Wang et al., 2021*) based on the STIRPAT model to determine the optimal emission reduction strategy for China. The flexibility of the STIRPAT model allows the consideration of more policy factors than the LMDI model, and the application of the STIRPAT model in conjunction with scenario simulation is an important advantage (*Vélez-Henao, Vivanco & Hernández-Riveros, 2019*; *Wei et al., 2023b*). The 14th Five-Year Plan and Outline of Vision 2035 of Heilongjiang mentions a specific development plan for the agricultural sector that provides useful data for future scenario analyses using the STIRPAT model. The prediction results indicated the extent of the impact of policy changes on agricultural carbon emissions in Heilongjiang Province.

Because of this, this study used Heilongjiang Province as an example and comprehensively analyzed agricultural carbon emissions from 1993 to 2020. Moreover, we identified the influencing factors and predicted future trends in different scenarios. Decomposition and prospective models of agriculture in Heilongjiang are universal in China. The choice of driving factors and construction of the STIRPAT model in the agricultural sector may provide a reference for other regions.

## MATERIALS & METHODS

### Study area

Heilongjiang Province is located in the northeast of China (43°26′–53°33′N, 121°11′–135°05′E) with an area of $45.25 \times 10^4$ km$^2$. Heilongjiang has a continental monsoon climate; the yearly average precipitation is 536 mm and the annual average temperature is 5.5 °C (*Wu et al., 2023b*; *Xi et al., 2023*). The gross output value of agriculture in Heilongjiang has developed from 29.99 billion Yuan in 1993 to 595.71 billion CNY in 2020, with an average annual growth rate of 11.7%, higher than the national growth rate of 9.82%. In 2020, the total grain production in Heilongjiang reached 75.4 Mt, accounting for 11.3% of the national grain production, which was the highest across the country. Commercial grain accounts for 38.7% of the production, which is the highest amount across the country. The mechanization rate of the major crops (plowing, planting, and harvesting) was above 98%, which was 30% higher than the national average. Rice, maize, and soybean are the main crops in Heilongjiang Province (*NBS, 2022*).

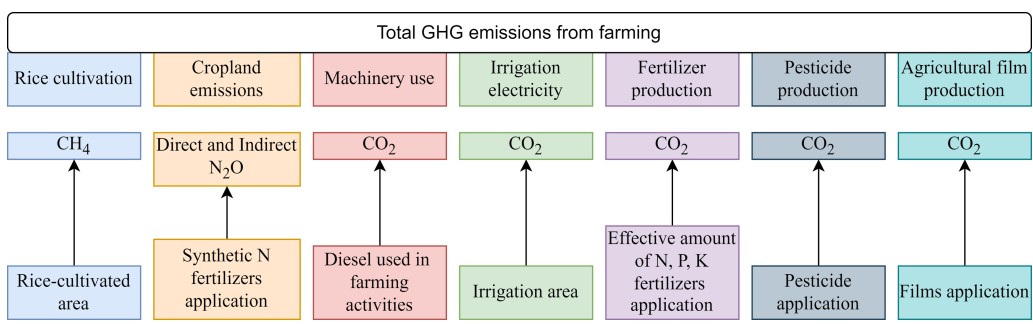

**Figure 1 Diagram of evaluation scope framework.**

**Table 1 Emission factors of agriculture inputs used in the evaluation.**

| Items | Emission factors | Unit | Sources |
|---|---|---|---|
| Rice cultivation | 168.0 | Kg/ha | *NDRC (2011)* |
| Machinery use | 0.8443 | tC/t | *Yan et al. (2022)* and *IPCC (2006)* |
| N fertilizer production | 2.116 | tC/t N | *Chen, Lu & Wang (2015)* |
| P fertilizer production | 0.636 | t C/t $P_2O_5$ | *Chen, Lu & Wang (2015)* |
| K fertilizer production | 0.14 | t C/t $K_2O$ | *Chen, Lu & Wang (2015)* |
| Compound fertilizer production | 1.77 | Kg C/kg | *Wu & He (2021)* |
| Irrigation electricity | 0.74 | t C/ ha | *Cheng et al. (2011)* |
| Pesticide production | 16.75 | kg C/kg | *Zhang et al. (2016)* |
| Agricultural film production | 22.7 | kg C/kg | *Huang et al. (2016)* |

## Carbon emissions from agriculture

In this study, the emissions from chemical fertilizers, pesticides, agricultural plastic films, machinery, and irrigation were analyzed.

$$C_{\text{total}} = \sum_{i}^{n} E_i \times EF_i \times GWP_i \qquad (1)$$

where $C_{\text{total}}$ is the total carbon emissions from agricultural activities, $E_i$ is the input amount of agricultural activity items, and n is the type of activity, which includes rice cultivation, cropland emissions, machinery use, irrigation electricity, fertilizer production, pesticide production, and agricultural film production (Fig. 1). The rice cultivated area; effective irrigated area; effective amounts of N, P, K, and compound fertilizers; amounts of pesticides; and agricultural film from 1993 to 2020 were all obtained from the Heilongjiang Statistical Yearbooks (*HLJBS, 1993–2020*). $EF_i$ is the carbon emission factor, $i$, as described in Table 1. $GWP_i$ is the warming potential of GHG emissions from agricultural activities, and $CH_4$ and $N_2O$ have a GWP of 27.2 and 273 times that of $CO_2$ on a 100-year time scale (*IPCC, 2021*).

Farmland $N_2O$ emissions due to nitrogen fertilizer application were obtained using Eqs. (2)–(5).

$$E_{N_2O} = \sum (N_{input} \times EF) \tag{2}$$

$$N_{input} = N_{direct} + N_{indirect} \tag{3}$$

$$N_{direct} = N_{fertilizer} \times EF_{direct} \tag{4}$$

$$N_{indirect} = N_{direct} \times 10\% \times 0.01 + N_{direct} \times 20\% \times 0.0075 \tag{5}$$

where $EF_{direct}$ is 0.0114 kgN$_2$O-N.

The Mann-Kendall (MK) test was used to analyze the time-series trends of carbon emissions (*Khambhammettu, 2005*; *Wang et al., 2022*). We first identified the years with the maximum carbon emissions of each agricultural activity and total carbon emissions and then conducted the MK test on changes in emissions from the maximum year to 2020 and the period 1993–2020, 2000–2020, and 2010–2020.

## Decomposition of agricultural carbon emissions factors

The Logarithmic Mean Divisia Index (LMDI) decomposition method was employed to analyze the evolution of agricultural GHG emissions in Heilongjiang Province from 1993 to 2020. We constructed a specific LMDI decomposition Eq. (6) for agricultural carbon emissions based on Kaya's identity (*Kaya, 1989*).

$$C = \frac{C}{Y} \times \frac{Y}{P_R} \times \frac{P_R}{P} \times P = CI \times EI \times SI \times DI \tag{6}$$

where $C$ represents the agricultural carbon emissions, $Y$ represents the gross value of agricultural production, $P_R$ represents the population working in agriculture, and P represents the total population. $CI = C/Y$ represents the intensity effect, which refers to carbon emissions per unit of the gross value of agricultural production. $EI = Y/P_R$ represents the economic effect, which refers to the share of unit labor input in the economic output of the agricultural sector; $SI = P_R/P$ represents the labor structure effect, which refers to the percentage of the total population working in the agricultural sector; and $DI = P$ represents the demographic effect. Agricultural carbon emissions in the past and t periods were set as $C_0$ and $C_t$, and C is the change in agricultural carbon emissions from the base period to period $t$. The effects of agricultural carbon emissions can be decomposed as follows.

$$\text{Intensity effect} \Delta CI = \sum \frac{C_t - C_0}{lnC_t - lnC_0} ln \frac{CI_t}{CI_0} \tag{7}$$

$$\text{Economic effect} \Delta EI = \sum \frac{C_t - C_0}{lnC_t - lnC_0} ln \frac{EI_t}{EI_0} \tag{8}$$

$$\text{Labor structure effect} \Delta SI = \sum \frac{C_t - C_0}{lnC_t - lnC_0} ln \frac{SI_t}{SI_0} \tag{9}$$

$$\text{Demographic effect}\, \Delta DI = \sum \frac{C_t - C_0}{lnC_t - lnC_0} ln\frac{DI_t}{DI_0} \tag{10}$$

$$\Delta C = C_t - C_0 = \Delta CI + \Delta EI + \Delta SI + \Delta DI. \tag{11}$$

## STIRPAT model for agricultural carbon emission trend prediction
### Standard STIRPAT model
Considering the three key driving forces: population, affluence, and technology based on the refined STIRPAT model (*York, Rosa & Dietz, 2003*) and factors chosen by *Qiu et al. (2022)* and *Yang et al. (2023)*. We used rural population (P), GDP per capita (A), intensity of agricultural carbon emissions (T), gross power of agricultural machinery (M), and number of technology patents (S) as specific driving factors. The agricultural carbon emissions (C) were predicted as follows:

$$C = aP^b \times A^c \times T^d \times M^f \times S^g \times e. \tag{12}$$

The STIRPAT model can convert all factors to natural logarithmic form as follows:

$$lnC = lna + blnP + clnA + dlnT + flnM + glnS + lne \tag{13}$$

where $a$ is a constant, and the driving factor coefficients of $b, c, d, f$, and $d$ refer to the percentage change in $C$ in response to a 1% change in the driving factor with others held constant.

### Partial correlation analysis
The partial correlation coefficients between rural population (P), GDP per capita (A), intensity of agriculture carbon emissions (T), gross power of agricultural machinery (M), and agricultural carbon emissions (C) are $-0.711$, $0.939$, $-0.868$, and $0.970$, respectively. The probabilities of significance (two-sided) tests are all below 1%, indicating that GDP per capita and gross power of agricultural machinery are significantly positively correlated with agricultural carbon emissions and that rural population, intensity of agricultural carbon emissions, and number of technology patents are significantly negatively correlated with agricultural carbon emissions.

### Modeling the influencing factors of carbon emission
Using the principal component analysis to reduce the dimensionality of lnP, lnA, lnT, lnM, and lnS, two principal components (F1 and F2) were extracted that explained 98.677% of the original variables (Table 2), and the significance test values were less than 0.01. Meanwhile, the relationship between FAC1, FAC2, and FAC3 and the original variables is obtained as

$$F1 = -0.462 \times lnP + 0.801 \times lnA - 0.721 \times lnT + 0.893 \times lnM \tag{14}$$

$$F2 = 0.882 \times lnP - 0.595 \times lnA + 0.675 \times lnT - 0.433 \times lnM \tag{15}$$
**Table 2  Principal component analysis of the total explained variance.**

| Component | | 1 | 2 | 3 | 4 |
|---|---|---|---|---|---|
| Initial eigenvalue | Eigenvalue | 3.741 | 0.206 | 0.048 | 0.005 |
| | Variance contribution rate % | 93.537 | 5.139 | 1.21 | 0.113 |
| | Accumulative contribution rate % | 93.537 | 98.677 | 99.887 | 100 |
| Extract sum of square loadings | Eigenvalue | 3.741 | 0.206 | | |
| | Variance contribution rate % | 93.537 | 5.139 | | |
| | Accumulative contribution rate % | 93.537 | 98.677 | | |
| Rotation sum of square loadings | Eigenvalue | 2.172 | 1.775 | | |
| | Variance contribution rate % | 54.299 | 44.378 | | |
| | Accumulative contribution rate % | 54.299 | 98.677 | | |

Using $lnI$ as the dependent variable and $F1$ and $F2$ as explanatory variables, second-order least squares regression analyses were performed to obtain the equations for the principal components and the dependent variable, $I$, as follows:

$$lnI = 1.438 \times F1 + 1.642 \times F2 - 3.595 R^2 = 0.970, F = 408.681, P < 0.001. \tag{16}$$

The Eq. (17) for the factors influencing agricultural carbon emissions in Heilongjiang from 1993 to 2020 can be transformed from (Eq. 14) and (Eq. 15).

$$lnI = 0.784 \times lnP + 0.175 \times lnA + 0.072 \times lnT + 0.574 \times lnM - 3.595. \tag{17}$$

Therefore, the multivariate nonlinear STIRPAT model for agricultural carbon emissions in Heilongjiang can be represented by Eq. (18). The trends of the simulations from the STIRPAT model and the estimated values were generally similar (Fig. 2), indicating that they can be used for the projection of agricultural carbon emissions.

$$I = e^{-3.595} \times P^{0.784} \times A^{0.175} \times T^{0.072} \times M^{0.574}. \tag{18}$$

## Scenario analysis setting

To analyze the impact of different development paths and policy preferences on agricultural carbon emissions in Heilongjiang, three scenarios based on the adjustments to the 14th five-year plan and outline of vision 2035 (https://www.ndrc.gov.cn/fggz/fzzlgh/dffzgh/202106/t20210628_1284318.html) were constructed: (i). Business as usual (BAU) scenario: This scenario follows the current emission reduction, economic development, and urbanization rates deployed in Heilongjiang's 14th Five-Year Plan and the outline of Vision 2035. Heilongjiang has achieved a machinery target with a 98% mechanization rate in farming. Therefore, the growth rate of machinery was set to zero. (ii). Intensity optimization scenario: This scenario maintains the economic development and urbanization rate in Heilongjiang's 14th Five-Year Plan, outlines Vision 2035 and improves carbon intensity by 20%. (iii). Slowdown in the economic development scenario: This scenario slows the economic growth and urbanization rate by 20% and improves the carbon intensity by 20%. Table 3 describes the parameters of the three scenarios.
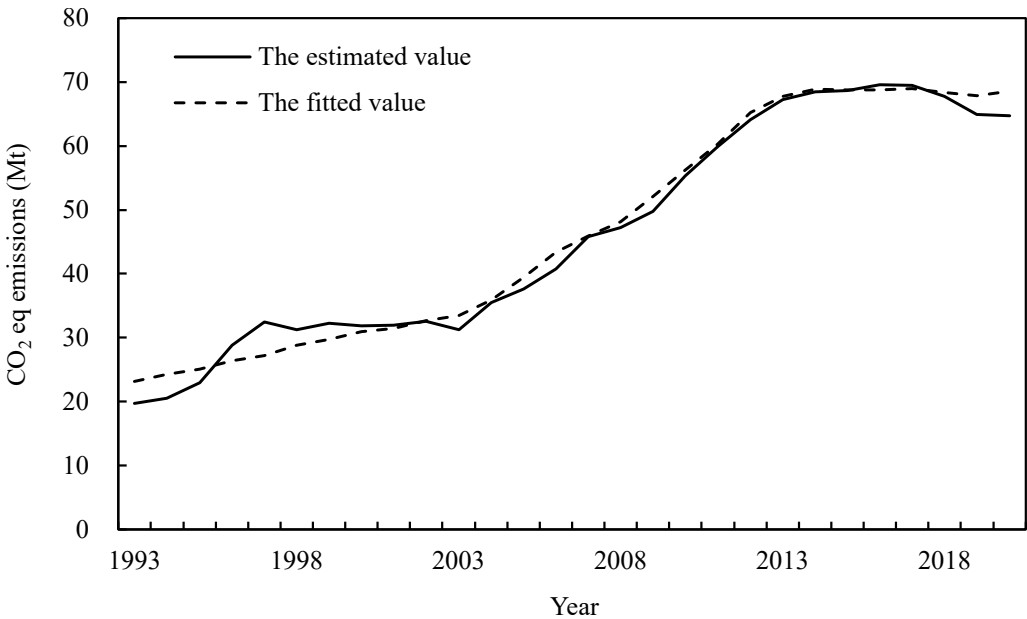

**Figure 2 Comparison between the estimated values and fitted values of carbon emissions.**

**Table 3 Parameters in each scenario.**

| Scenarios | Period | Growth rate % | | | |
|---|---|---|---|---|---|
| | | **P** | **A** | **T** | **M** |
| Business as usual scenario | 2020–2025 | −1.35 | 5.50 | −4.84 | 0.00 |
| | 2026–2030 | −1.08 | 4.50 | −4.50 | 0.00 |
| Intensity optimization scenario | 2020–2025 | −1.35 | 5.50 | −5.80 | 0.00 |
| | 2026–2030 | −1.08 | 4.50 | −5.40 | 0.00 |
| Slowdown in economic develop-ment scenario | 2020–2025 | −1.08 | 4.40 | −5.80 | 0.00 |
| | 2026–2030 | −0.86 | 3.60 | −5.40 | 0.00 |

# RESULTS

## Agricultural carbon emissions

Carbon emissions from agricultural activities in Heilongjiang Province showed an overall increasing trend, peaking in 2016 at 69.6 Mt $CO_2$-eq before declining to 64.7 Mt $CO_2$-eq in 2020 (Fig. 3). The process can be divided into four stages. The first stage was from 1993 to 1997, which was the initial stage of increase with an average annual growth rate of carbon emissions of 13.3%. Subsequently, there was a stationary phase from 1998 to 2003, followed by a persistent increase from 2004 to 2016, during which the average annual growth rate of carbon emissions was 5.8%. After 2016, agricultural carbon emissions gradually decreased in the fourth stage. The trends in carbon emissions per unit hectare were similar to those of the total agricultural emissions, which increased from 2.3 to 4.3 t $CO_2$-eq · ha$^{-1}$ between 1993 and 2020. The two peaks were 4.7 t $CO_2$-eq · ha$^{-1}$, which occurred in 2013 and 2017

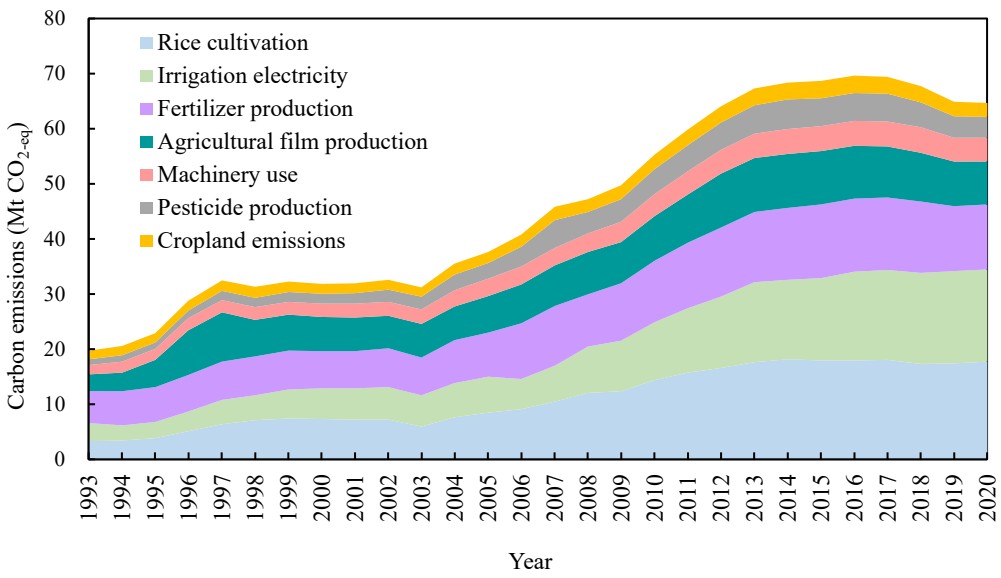

**Figure 3** Carbon emissions from agricultural activities in Heilongjiang Province, 1993–2020.

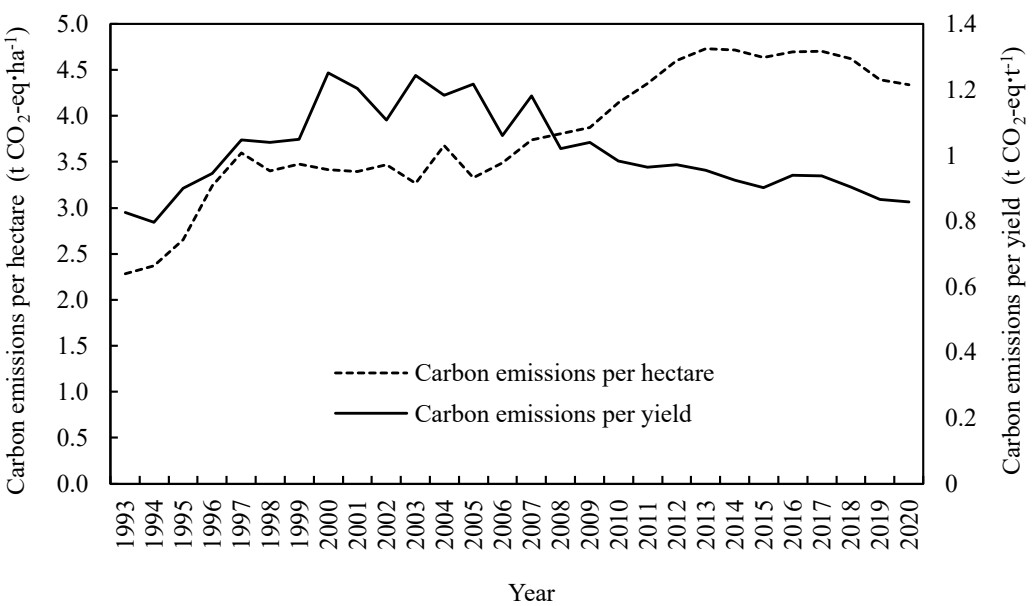

**Figure 4** Carbon emission intensity of agricultural activities in Heilongjiang Province, 1993–2020.

(Fig. 4). On the other hand, carbon emissions per unit yield showed a steady downward trend after 2008. The level was comparable in 2020 (0.85 t $CO_2$-eq $\cdot$ $t^{-1}$) to the 1993 (0.83 t $CO_2$-eq $\cdot$ $t^{-1}$).

Regarding the composition of agricultural carbon emissions, rice cultivation accounted for the largest proportion (27.3% in 2020), surpassing the emissions from fertilizer

**Table 4  MK test results for carbon emissions of agricultural acitvities.**

| Agricultural activities | Z value | | | Maximum year | Z value |
|---|---|---|---|---|---|
| | 1993–2020 | 2000–2020 | 2010–2020 | | Maximum year-2020 |
| Rice cultivation | 6.3023*** | 4.9221*** | 1.5570 | 2014 | −1.2015 |
| Irrigation electricity | 7.0163*** | 5.7729*** | 3.8925*** | 2019 | – |
| Fertilizer production | 5.9072*** | 4.6201*** | 0.6404 | 2015 | −2.6301*** |
| Agricultural film production | 3.9711*** | 3.6538*** | −1.5570 | 2013 | −3.3404*** |
| Machinery use | 6.8371*** | 5.3775*** | 1.9521* | 2018 | – |
| Pesticide production | 5.7096*** | 3.6538*** | −0.7785 | 2017 | – |
| Cropland emissions | 5.3145*** | 4.1370*** | −0.4671 | 2015 | −2.6301*** |
| Total emissions | 6.2628*** | 5.0429*** | 1.4013 | 2016 | −2.2045 |

**Notes.**
Positive Z values correspond to increasing trends in carbon emissions, and vice versa; *, **, *** denotes that the trend is significant at 0.1, 0.05 and 0.01 level, respectively; – indicates that the maximum value existed for less than five years.

production in 1998. Carbon emissions from fertilizer production peaked at 13.4 Mt $CO_2$-eq in 2015, before decreasing annually to 11.8 Mt $CO_2$-eq in 2020. Irrigation electricity surpassed fertilizer production to become the second largest source of carbon emissions in 2010. Carbon emissions from irrigation electricity were 16.7 Mt $CO_2$-eq in 2020, which is 5.3 times higher than that in 1993. Carbon emissions from agricultural film production, machinery use, pesticide production, and croplands were 7.9, 4.3, 3.7, and 2.6 Mt $CO_2$-eq, respectively, in 2020. Trend analysis of the MK test (Table 4) showed that all agricultural activities, except pesticide production and cropland emissions, exhibited significantly increasing trends in carbon emissions during the entire study period (1993–2020) and shorter periods (2000–2020 and 2010–2020). However, for pesticide production and cropland emissions, the trends were not significant in the later periods. Irrigation electricity and machinery use have not yet demonstrated significant carbon emission reductions, indicating their mitigation potential.

## LMDI decomposition

The decomposition results (Table 5) showed that the economic and labor structure effects were the two factors that drove the overall increase in agricultural carbon emissions overall during 1993–2020. The economic effect was the key factor, with 96.84 Mt of agricultural carbon emissions, which was 16 times higher than the labor structure effect. In terms of factors inhibiting agricultural carbon emissions, the agricultural carbon emission intensity effect and demographic effect contributed –46.66 and –11.10 Mt of agricultural carbon emissions in reducing agricultural carbon emissions in Heilongjiang. Economic and intensity effects play decisive roles in determining the overall amount of agricultural carbon emissions. The economic effect showed the strongest increase during 2010–2013, while the maximum total agricultural carbon emissions did not occur in this period because of the inhibitory effect of the intensity factor. After 2016, only the economic effect continued to increase, and the total effect decreased the agricultural carbon emissions.

**Table 5 The decomposition results of the driving mechanism for agricultural carbon emissions in Heilongjiang (Mt).**

| Year | Intensity effect | Economic effect | Labor structure effect | Demographic effect | Total effect |
|---|---|---|---|---|---|
| 1993–1994 | −4.56 | 5.42 | −0.23 | 0.18 | 0.81 |
| 1994–1995 | 1.66 | 0.29 | 0.29 | 0.17 | 2.40 |
| 1995–1996 | 3.01 | 2.05 | 0.58 | 0.19 | 5.84 |
| 1996–1997 | 3.87 | −15.70 | 15.34 | 0.19 | 3.70 |
| 1997–1998 | 1.67 | −0.39 | −2.67 | 0.19 | −1.21 |
| 1998–1999 | 4.25 | −2.64 | −0.82 | 0.16 | 0.95 |
| 1999–2000 | 3.11 | −3.45 | −0.15 | 0.13 | −0.36 |
| 2000–2001 | −2.39 | 2.52 | −0.10 | 0.03 | 0.06 |
| 2001–2002 | −2.14 | 2.65 | 0.13 | 0.02 | 0.66 |
| 2002–2003 | −1.97 | 1.09 | −0.50 | 0.02 | −1.36 |
| 2003–2004 | −1.43 | 7.04 | −1.34 | 0.02 | 4.28 |
| 2004–2005 | −2.59 | 5.23 | −0.52 | 0.03 | 2.14 |
| 2005–2006 | −1.34 | 4.87 | −0.43 | 0.03 | 3.13 |
| 2006–2007 | 4.23 | 1.75 | −0.93 | 0.01 | 5.07 |
| 2007–2008 | −4.64 | 5.78 | 0.18 | 0.01 | 1.34 |
| 2008–2009 | −1.54 | 3.67 | 0.42 | 0.01 | 2.56 |
| 2009–2010 | −0.56 | 6.67 | −0.60 | 0.10 | 5.60 |
| 2010–2011 | −9.62 | 14.12 | 0.79 | −0.77 | 4.51 |
| 2011–2012 | −11.17 | 16.36 | 0.00 | −0.96 | 4.23 |
| 2012–2013 | −10.45 | 13.69 | 0.97 | −1.03 | 3.18 |
| 2013–2014 | −2.78 | 5.87 | −0.86 | −1.08 | 1.15 |
| 2014–2015 | 1.97 | −1.17 | 0.94 | −1.52 | 0.23 |
| 2015–2016 | 1.61 | 0.43 | 0.22 | −1.31 | 0.96 |
| 2016–2017 | −4.94 | 6.18 | −0.10 | −1.30 | −0.15 |
| 2017–2018 | −3.40 | 3.57 | −0.37 | −1.47 | −1.67 |
| 2018–2019 | −3.48 | 1.50 | 0.55 | −1.45 | −2.88 |
| 2019–2020 | −3.06 | 9.45 | −4.88 | −1.69 | −0.18 |
| Total | −46.66 | 96.84 | 5.92 | −11.10 | 44.99 |
| Percentage (%) | −103.70 | 215.23 | 13.15 | −24.68 | 100.00 |

## Scenario analysis based on the STIRPAT model

As shown in Eq. (18), the rural population has a negative effect, with a coefficient of 0.551; that is, for every 1% increase in the rural population, the total agricultural carbon emissions would increase by 0.551%. Similarly, for every 1% increase in the GDP per capita of rural residents, gross power of agricultural machinery, and the number of technology patents, agricultural carbon emissions increased by 0.023%, 0.255%, and 0.225%, respectively. Only the intensity of agricultural carbon emissions has a positive effect. For every 1% reduction in the intensity of agricultural carbon emissions, total agricultural carbon emissions decreased by 0.005%.

The predicted agricultural carbon emissions in Heilongjiang under the three scenarios show decreasing trends between 2020 and 2030 (Fig. 5). The reductions in 2030 for the BAU, intensity optimization, and a slowdown in economic development scenarios compared to

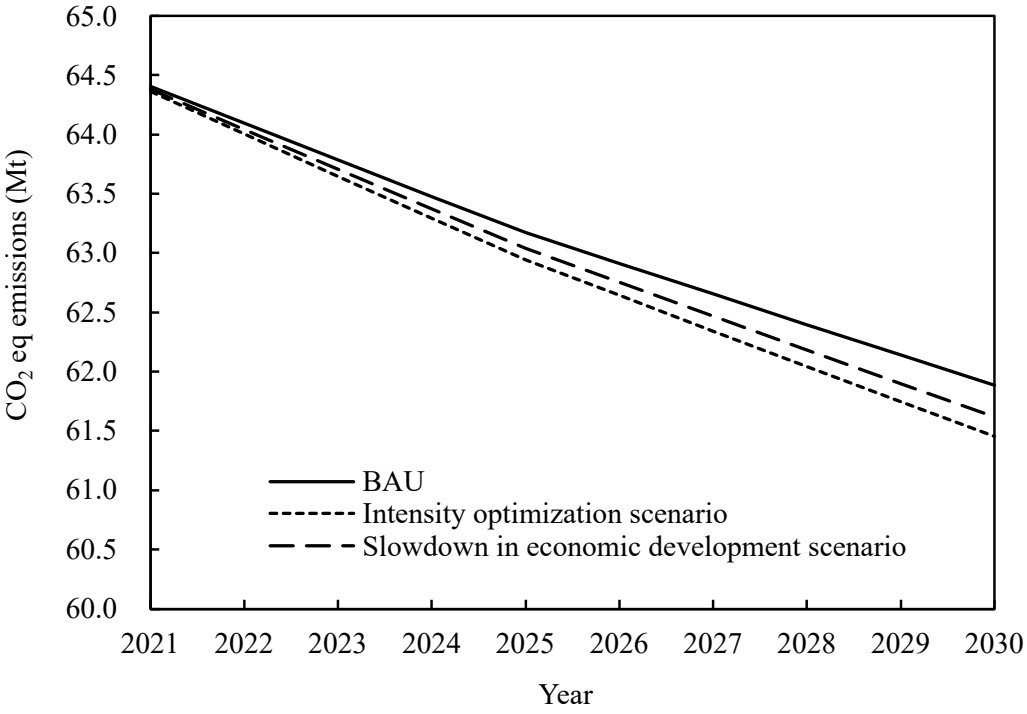

**Figure 5 Agricultural carbon emission trends in Heilongjiang under different scenarios, 2021–2030.**

agricultural carbon emissions in 2021 are predicted to be −3.92%, −4.52%, and −4.29%, respectively. A 20% improvement in intensity can produce a $45.9 \times 10^4$ t carbon emission reduction by 2030. Slowing economic growth would offset a few advantages (0.23%) from intensity improvement to agricultural carbon emission control. Overall, the three scenarios suggest a limited reduction in agricultural carbon emissions over the next decade, indicating a plateau in emission control.

## DISCUSSION

### Driving mechanism of agricultural carbon emissions in Heilongjiang Province

Macro policies and the economic climate impact the trend in carbon emissions from agriculture (Fig. 6). During the first phase (1993–1997), when China's market economy began to start and non-compound fertilizer value-added tax exemptions and agricultural subsidies were implemented (*MOF, 1995*), the rise in agricultural carbon emissions was matched by increased agricultural production. The decomposition results show that the economic effect had a decreasing effect on promoting agricultural carbon emissions, and the effects of intensity, labor structure, and demographics promoted agricultural carbon emissions at this stage. The main features of the first stage were low efficiency and fluctuating economic income.

The second stage (1998–2003) was the background of the Asian financial crisis, in which the amount of agricultural film used by farmers was significantly reduced to reduce the

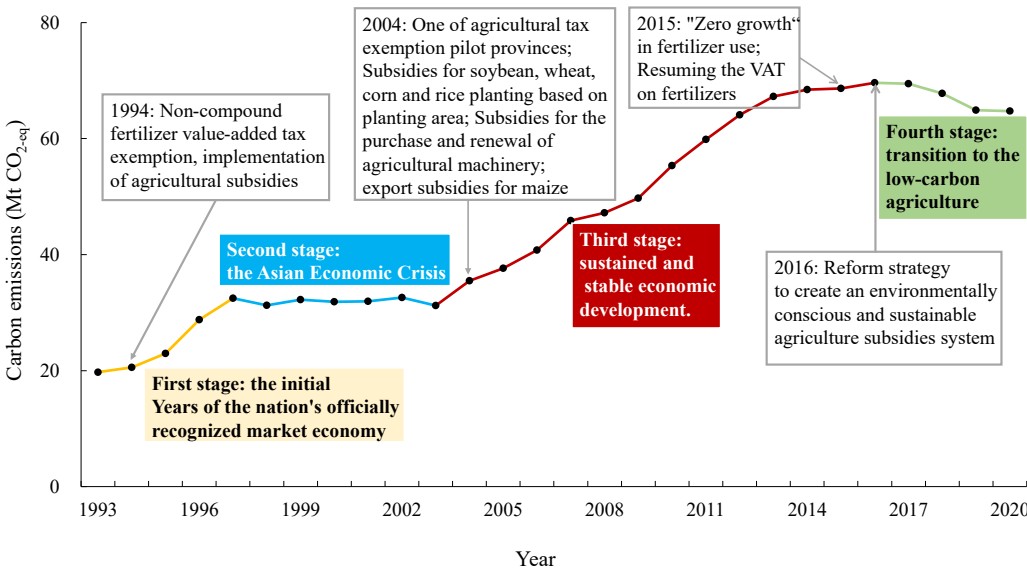

**Figure 6** Agricultural policies and trends in carbon emissions.

cost of agricultural inputs (*Xie, Cai & Huang, 2009*). However, carbon emissions from electricity used for irrigation caused by the expansion of rice cultivation increased; thus, the overall agricultural carbon emissions did not change significantly. The intensity effect promoted agricultural carbon emissions during the second stage, whereas the transition of the increasing effect from economic factors was a signal of rapid agricultural development in the next stage.

Various agricultural policies have been implemented in Heilongjiang since 2004. Heilongjiang abolished its agricultural tax two years ahead of schedule as a prototype province (*SCIO, 2004*). Moreover, there were subsidies for soybean, wheat, corn, and rice planting based on the planting area and the purchase and renewal of agricultural machinery for farmers in Heilongjiang. Consequently, agricultural carbon emissions in Heilongjiang increased from 2004 to 2016 with China's economic boom. During the third stage, the intensity and demographic effects curbed agricultural carbon emissions by −36.87 and −6.47, while the economic effect increased agricultural carbon emissions by 77.26. The role of the intensity effect in reducing emissions is evident in rapid economic development. The onset of the demographic effect suppressing agricultural carbon emissions (in 2010) and the timing of the population decline in Heilongjiang are the same (*NBS, 2022*), indicating that the decline in population plays a positive role in controlling agricultural carbon emissions in Heilongjiang.

China advocated "zero growth" in fertilizer use and green development in 2015 (*CPMMOAC, 2015*). Our results show that carbon emissions from fertilizer production in Heilongjiang have decreased annually since 2015. In 2016, the government proposed a reform plan to create an environmentally conscious and green agricultural subsidy system. These regulations marked the beginning of the agricultural sector's shift toward

low-carbon practices. Only the economic effect promoted agricultural carbon emissions after peaking in 2016, indicating the high efficiency and better labor structure of agriculture in Heilongjiang.

As indicated above, in addition to its direct relationship with the scale of production, agricultural carbon emissions are closely related to policy guidance, indicating the important role of policymaking in the future for further emission reduction goals.

### Future perspectives and policy implications

Heilongjiang Province's agricultural carbon emissions peaked in 2016, comparable to China's agricultural carbon emissions, which peaked in 2015 (*Wu et al., 2023a*). The average annual decline rate was −1.80% after peaking in Heilongjiang in 2016 and 2020, which was similar to the national level of −1.56% (*Wu et al., 2023a*). Based on the anticipated scenarios, it is projected that agricultural carbon emissions in Heilongjiang will continue to decrease gradually over the forthcoming decade, indicating the constraints of existing means of emission reduction. Among the influencing factors, an increase in the number of technology patents did not inhibit carbon emissions, which differs from the results of other Chinese provincial studies (*Li, Liu & Li, 2015*; *Yang et al., 2023*). This may indicate that technological upgrading has not yet played a significant role in reducing agricultural emissions. Moreover, it implies that more can be done in this area in the future to help reduce agricultural carbon emissions.

Through a review of historical and current agricultural emission reduction policies, it is evident that the approach of optimizing cropping systems to control carbon emissions has not been explicitly mentioned. Heilongjiang Province, which has the largest rice cultivation area, contributes significantly to methane and carbon emissions from irrigation electricity, accounting for 53% of the total emissions in 2020. Reducing the rice cultivation area appropriately, encouraging rotation between dryland crops and rice, or guiding farmers toward the cultivation of upland rice varieties could contribute significantly to mitigating overall agricultural carbon emissions. The GHG emission intensity of upland rice is only 23%–54% of that of ordinary rice paddies (*Nishimura et al., 2011*; *Sun et al., 2020*; *Weller et al., 2016*).

### Limitations

This study had certain limitations. First, owing to the availability of data, the driving factors used in the STIRPAT model may not be sufficiently comprehensive. Second, agricultural carbon emissions were accounted for based on agricultural activities as recorded in statistical yearbooks. New insights can be gained by accounting for carbon emissions according to specific crop types.

## CONCLUSIONS

In this study, we adopted a decomposition analysis and trend prediction to examine the relationship between grain production and agricultural carbon emissions in Heilongjiang Province from 1993 to 2030. Decomposition analysis indicated that the economic effect was the key factor driving the overall increase in agricultural carbon emissions overall during

1993–2020. In contrast, the intensity effect is the major factor controlling agricultural carbon emissions. Population reduction is favorable for achieving agricultural emission reductions in Heilongjiang. In 2020, $CH_4$ emissions from rice cultivation and carbon emissions from irrigation electricity were the main components of agricultural carbon emissions in Heilongjiang Province, accounting for 27.3% and 25.9%, respectively. Carbon emissions from irrigation electricity and machinery did not peak in the trend analysis. The agricultural carbon emissions in Heilongjiang Province reached a peak in 2016, with the total emissions being 3.5 times higher than those in 1993. According to scenario predictions, agricultural carbon emission reductions over the next decade would be $-3.92\%$ to $-4.52\%$ yet achieving carbon neutrality remains a significant challenge. Overall, the methods and conclusions offered in Heilongjiang case study not only provide useful implications in analyzing the factors that drives carbon emissions at the provincial level but also propose measures to formulate low-carbon plans for policymakers. Such a systematic analysis an be extended to other regions by considering their realities.

## ACKNOWLEDGEMENTS

We thank the associate editor and the reviewers for their useful feedback that improved this paper.

### Funding

This study was supported by the Chinese National Natural Science Foundation No. 41301316, Joint Funds of the Natural Science Foundation of Heilongjiang Province of China (LH2021C025) and the ''Young Talents'' Project of Northeast Agricultural University (2022). The funders had no role in study design, data collection and analysis, decision to publish, or preparation of the manuscript.

### Grant Disclosures

The following grant information was disclosed by the authors:
The Chinese National Natural Science Foundation: 41301316.
Joint Funds of the Natural Science Foundation of Heilongjiang Province of China: LH2021C025.
The''Young Talents'' Project of Northeast Agricultural University (2022).

### Competing Interests

The authors declare there are no competing interests.

### Author Contributions

- Luyang Zhang conceived and designed the experiments, performed the experiments, analyzed the data, prepared figures and/or tables, authored or reviewed drafts of the article, and approved the final draft.
- Junyan Chu performed the experiments, analyzed the data, authored or reviewed drafts of the article, and approved the final draft.

- Haiyang You performed the experiments, analyzed the data, authored or reviewed drafts of the article, and approved the final draft.
- Zhihua Liu conceived and designed the experiments, authored or reviewed drafts of the article, and approved the final draft.

## Data Availability

The data is available in the Supplemental File and came from the Yearbook database at China National Knowledge Infrastructure (CNKI):

https://oversea.cnki.net/KNavi/YearbookDetail?pcode=CYFD&pykm=YHLTN&bh=&uniplatform=OVERSEA&language=en.

The search parameters used: rice cultivated area 水稻种植面积; diesel used in farming 柴油用量; irrigation area 灌溉有效面积; N/P/K-fertilizers application 氮、磷、钾肥; pesticide 农药; films 农膜。

## Supplemental Information

Supplemental information for this article can be found online at http://dx.doi.org/10.7717/peerj.17856#supplemental-information.

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
