# Peer review of "Decomposition and scenario analysis of agricultural carbon emissions in Heilongjiang, China"

_PeerJ, doi:10.7717/peerj.17856_

## Round 0.1 · original submission · Major Revisions

Based on the anomynous reviews, your manuscript required major revision before it can be accepted. Please try to submit the revised manuscript and reply within three weeks.

Reviewer 1 ·

Basic reporting

Zhang et al. studied and decomposed the driving factors of agricultural carbon emissions in Heilongjiang Province and conducted scenario analysis. The study is based on a large amount of work, but is currently not well written.

Introduction
Line 38: This sentence is too confusing. Maybe you want to express that human activities have strongly affected climate change.

Lines 42-44: Literature should be cited.

Lines 60-69: What do you want to illustrate in this paragraph, and what research gaps have been raised? For carbon emission units, some places use equivalents, which may be less common.
It is highly recommended to rewrite the introduction. The description of the research gap is insufficient, and now it is only a list of research methods. The research gap you proposed is the influencing factors of carbon emissions in Heilongjiang after 2013. It is too simple.

Materials & Methods
Lines 138: What is **

Lines 136-140: Regarding formulas 6 and 7, you did not decompose them according to formula 6, so why did you give formula 6? Energy consumption (E) and production (Y), as well as population and GDP, do not mean the same thing. You can consider decomposing one more industrial structure, agricultural GDP/total GDP, population engaged in agriculture/agricultural GDP.

Lines 156: I don't think it can be written as total impact, it's just the change in carbon emissions.

Lines 158: Check ()

Results
Lines 184-191: You might consider a trend analysis of carbon emissions (Mann-Kendall……).

Lines 214: Partial correlation analysis needs to be written clearly in the method part.

Lines 223: Why are there formulas in the results?

References
Please check the format of your references carefully.

Experimental design

Please refer to the first area.

Validity of the findings

Please refer to the first area.

Additional comments

No

Reviewer 2 ·

Basic reporting

This manuscript estimated agricultural carbon emissions in Heilongjiang Province, China, from 1993 to 2020 using the Logarithmic Mean Divisia Index (LMDI) and the STIRPAT. The study is comprehensive and provides valuable insights into the factors influencing emissions and potential future trends.

Experimental design

I appreciate the use of the LMDI and STIRPAT models to identify and project the driving forces of these emissions.

Validity of the findings

The findings of this manuscript could have some contributions to green development of agriculture.

Additional comments

However, the manuscript should be greatly improved before it could be considered for publication. In addition, the English writing should be greatly improved.
There are some specific suggestions.
Abstract:
Line 18: “China’s” should be revised. It is not proper to use ‘s in an academic article. “has witnessed” also should be revised.
Line 23 delve deeper?
Line 27: There's an extra space.
There is no clear information about predicted emissions of Heilongjiang, it should describe here as an important section of the paper.
Introduction
I suggest the author rewrite the introduction. There is too much Chinese English.
Line 44 between should be deleted.
Line 46 there a redundant “.” before (IPCC 2022). In addition, the citations seem not be in the proper format.
Line 50 “China’s” should be revised.
Line 54 “pathway” do you mean “method” or “policy”?
Line 52 mentioned the national determined contributions, how about the province contributions of Heilongjiang? It could be elaborated in more detail as the useful background information.
What are the links between STIRPAT and LMDI? The applications in agriculture and the reasons for the choice of the two models for the article should be elaborated in more detail in the Introduction section.
Line 60: “have well established” is there a “be”?
Line 72: “Logarithmic Mean Divisa Index” the initials should all be capitalized.
Materials & Methods
Line 102-103:There should be a space between the number and the unit.
Line 104-111: Provide citations in the text.
Line 116:which i should be subscripted.
Line 138:The equation number is missing.
Results
Carbon intensity per unit area or yield should be analyzed in the Agricultural Carbon Emissions section.
Figure 4: Projected data and historical assessment trends did not match. As shown in Figure 4, the emissions are between 6800 and 6850, while the line 185 says the emissions are 64.7 Mt in 2020, and the study predicts a declining trend. The conflict here needs to be corrected. The units of the y-axis of Figure 4 are also not the same as those in the previous figures.
Line 223: STIRPAT model is not a very appropriate title.
Discussion
Line 296: It would be better with some concluding statements at the end of Driving mechanism of agricultural carbon emissions in Heilongjiang province section.
Line 304: Consider discussing more with the STIRPAT model analysis in the future perspectives part.
The discussion could be expanded to compare the findings with other regions in China or globally, highlighting any similarities or differences.
What are the limitations of the study? They should be discussed in this section.
Conclusions
Line 311 in this paper, should be in this study.
Line 315: I believe it should be “key factor driving” rather than“key factor drove”.
Line 316: It should be“intensity effect”.
Overall, the conclusion section provides a clear summary of the study's findings but could have been enhanced by including more specific proportions or data in describing the main components. In addition, the author should provide more information about the implications of this study.

References
The format of the references needs to be adjusted, such as Line 374: “Statistics NBo”, Line 392 and 407: “CO2”, Line 437: format of journal name is not uniform.

Figure
In Figure 1, some text is outside the intended boundaries. Please check it.
Table
Line 1: There should be a space between the Table and 1. In Table 1, these tC/t P205 are more appropriately placed in one line; two lines tend to be ambiguous. Similar to t C/t K2O.
The reading experience of Table 4 is not good, please try to adjust it.

Annotated reviews are not available for download in order to protect the identity of reviewers who chose to remain anonymous.

---

## Round 0.2 · accepted · Accept

I have evaluated your revisions and the article is now Acceptable. Thank you again for your contribution to the PeerJ.